# Results of Anterior Cruciate Ligament Avulsion Fracture by Treatment Using Bioabsorbable Nails in Children and Adolescents

**DOI:** 10.3390/children9121897

**Published:** 2022-12-02

**Authors:** Łukasz Wiktor, Ryszard Tomaszewski

**Affiliations:** 1Department of Trauma and Orthopaedic Surgery, Upper Silesian Children’s Health Centre, 40-752 Katowice, Poland; 2Department of Trauma and Orthopedic Surgery, ZSM Hospital, 41-500 Chorzów, Poland; 3Faculty of Science and Technology, Institute of Biomedical Engineering, University of Silesia in Katowice, 40-007 Katowice, Poland

**Keywords:** tibial eminence, fracture, children, bioabsorbable, anterior cruciate ligament, ACL

## Abstract

(1) Background: Anterior cruciate ligament avulsion fractures are characteristic for skeletally immature patients, and appropriate treatment is currently debated in the literature. The study aimed to evaluate the clinical and functional outcomes in patients with tibial eminence fractures treated with bioabsorbable nails in one orthopedic clinic. (2) Methods: After retrospective evaluation, we found 17 patients with tibial eminence fractures treated in orthopedic departments between January 2013 and July 2022 using bioabsorbable fixation nails. The study group comprised 12 boys and five girls aged 5 to 15.2 (average 10.1). The mean follow-up was 28 months. We diagnosed five type II fractures, ten type III fractures, and two type IV fractures according to Meyers–McKeever classification. (3) Results: We obtained a high healing rate—17 patients with the complete union on the control radiographs. We diagnosed two cases of malunion, of which one required revision surgery. Only one patient showed a slight anterior knee laxity. The treatment effect at follow-up was assessed using the Lysholm Knee Score and IKDC Score. The median Lysholm Score was 96.64 (SD 4.54), and the median IKDC Score was 84.64 (SD 3.10), which were both excellent results. (4) Conclusions: Based on our results, surgery using bioabsorbable devices for type II, III, and IV tibial eminence fractures in young individuals is an effective alternative, allowing good outcomes and restoring proper knee stability. The crucial factor for a good effect is a stable fracture fixation. Arthroscopic surgery gives good outcomes with minimal invasion. It is important not to prolong the attempts of arthroscopic reduction and to perform the open reduction to shorten the procedure’s time and avoid complications.

## 1. Introduction

Avulsion fractures of the tibial insertion of the anterior cruciate ligament (ACL) are characteristic for skeletally immature children, most commonly between 8 and 14, with an annual incidence of 3 per 100,000 [1,2,3]. They usually result from sports-related activities [4,5]. In 1959, Meyers and McKeever described the original and most widely used classification system that divides tibial eminence fractures into three types [6]. In type I, the fragment is minimally displaced and typically treated non-surgically. In type II, the anterior part of the eminence is elevated but with an intact hinge posterior. Type III is completely displaced. In 1977, Zaricznyj B. introduced type IV fracture, which is displaced and comminuted [7]. Type II fractures can be treated with a closed reduction by knee hyperextension, while fractures with significant displacement are indicated for surgery [8]. Types III and IV are always treated with surgical reduction and internal fixation [9]. Surgical management is demanding because correction of ACL tension is crucial for knee stability. Malreduction may result in a reduced range of motion or knee instability. Closeness to the proximal tibial physis is also challenging. The recent surgical approach includes internal fixation of the fracture by arthroscopy—arthroscopic reduction and internal fixation (ARIF) [10,11]. Surgical management involving open reduction and internal fixation (ORIF) lately is dedicated to cases with arthroscopic reduction failure [9]. Internal fixation can be performed using different types of orthopedic implants such as K-wires, screws, suture material, and bioabsorbable nails or screws [12,13]. The main advantage of bioabsorbable devices is that the implant does not need to be removed [14,15]. On the other hand, the use of bioabsorbable implants is not without drawbacks. The most frequently reported are implant destabilization, implant breakage, uneven resorption, which can lead to implant back-out, and synovitis related to the host response to the polymer’s biodegradation. These devices are also invisible on X-rays, which hinders the follow up evaluation [15]. ARIF or ORIF with a metal screw fixation provides good biomechanical stability. However, the screw usually crosses the proximal tibial growth plate and requires removal [4,16]. This study aimed to evaluate the clinical and functional outcomes in patients with tibial eminence fractures treated with bioabsorbable nails.

## 2. Materials and Methods

We retrospectively evaluated the outcome of tibial eminence fracture in children and adolescents treated at our department. We used the STROBE protocol for retrospective observational studies [17]. The inclusion criteria were: (1) age under 18 y.o., (2) displaced tibial eminence fracture (type II, III, IV Meyers–McKeever), (3) surgical treatment with bioabsorbable pins, (4) follow-up in at least six months. We found 17 patients who met the inclusion criteria between January 2013 and July 2022. All patients were qualified for the surgery based on the antero–posterior and lateral X-rays of the affected knee, followed by a computed tomography (CT) examination. Based on CT scans, the fracture morphology was evaluated, and the classification according to Meyers–McKeever was assessed. Additional magnetic resonance imaging (MRI) was performed only on a few patients before the surgery in our study. MRI can be used, especially in diagnosing concomitant knee injuries such as meniscus tears or chondral damage. An overview of the study group is presented in Table 1.

### 2.1. Surgical Technique

All patients were treated using 2.4 × 25 mm SmartNails (ConMed, Linvatec, Tampere, Finland), which are osteochondral fixation nails made from poly-96L/4D-lactide copolymer [18,19].

The patient was placed in a supine position with an affected limb in a dedicated arthroscopic leg holder with the knee flexed to 90 degrees. A tourniquet was placed in all cases. As the first stage for all patients, diagnostic knee arthroscopy was performed with standard anterolateral and anteromedial portals. The intra-articular hematoma was rinsed and combined with joint debridement before assessing the fracture. An arthroscopic shaver was used to debride the fracture side from hematoma, small loose bone fragments, and any soft tissue interposing fracture reduction. We found it helpful to use a sharp-ended tibial guide from an ACL reconstruction set to reduce eminence fracture and hold it on the right site. Using above mentioned guide arm—Figure 1, it is effortless to stabilize tibial eminence with K-wire by retrograde drilling, temporarily. Moreover, it reduces the risk of further fracture damage. The final stabilization could be made if a satisfactory arthroscopic reduction was gained. In case of arthroscopic approach failure or the multi-fragmented nature of the fracture, a mini-arthrotomy was made for direct fracture visualization (medial parapatellar arthrotomy approximately 3–4 cm in length). Once the fragment was reduced using an accessory superomedial portal (approximately 1 cm in length) or a mini-arthrotomy performed earlier, a dedicated guide handle was inserted into the joint on top of the fragment for the SmartNail implantation. After establishing the alignment of the assembly using a 2.4 mm diameter drill, the canal was drilled through the fragment into the solid bone until reaching the desired depth. The procedure usually was repeated by inserting a second implant at an angle to the first one to secure the fixation. Due to the implants, we used a radiolucent and fracture reduction was confirmed under direct visualization and intraoperative fluoroscopy. The number of bioabsorbable pins used in the particular cases depended on the fracture type and the surgeon’s experience. The surgical procedure diagram is shown in Figure 2.

Postoperatively, the affected knee was immobilized in a plaster splint for four weeks. The patient was allowed to walk with crutches with partial weight bearing. All patients stayed under rehabilitation outpatient control. After the cast was removed, hinged orthosis was applied, gradually increasing the range of motion. An example of a patient with a type II fracture being treated with an ARIF procedure (fixation with 2 SmartNails) is presented in Figure 3.

### 2.2. Clinical Follow-Up

Standard antero–posterior and lateral X-rays were taken to confirm proper intercondylar eminence bone healing at outpatient clinic control. Six months after the surgery, active knee range of motion was measured with a goniometer. A comprehensive knee examination with regard to the unaffected knee included an anterior drawer test, Lachman test, and pivot shift test. We estimated the results of the Lachman test and anterior drawer test, if positive, as grade I with an anterior tibial translation of 1–5 mm, grade II with a translation of 6–10 mm, and grade III with laxity more than 10 mm. For the pivot-shift test, we assumed grade 0 as normal, with no reduction or shifting, grade I with mild sliding, grade II with moderate tibial shifting while reducing, and grade III with the tibia starting subluxed and reducing with flexion, causing a characteristic clunk [20]. The functional outcome was evaluated with the Lysholm knee score [21,22]. We assessed the score as excellent: 95–100 points, good: 84–94 points, fair: 65–83, and poor <65 points [23] and the International Knee Documentation Committee (IKDC) score [24]. At a follow-up appointment, patients were asked to compare their activity levels before and after treatment and qualification as the same or lower. Patients remained under the orthopedic clinic control despite ending the treatment for further follow-up.

All procedures in the study were in accordance with the ethical standards of the institutional and/or national research committee and with the 1964 Helsinki declaration and its later amendments or comparable ethical standards.

## 3. Results

The study group comprised 12 boys and five girls aged 5 to 15.2 (average 10.1) years. The mean follow-up was 28 months (9 to 102; SD 21.9). The right knee was affected eight times, and the left nine times. We diagnosed five type II fractures, ten type III fractures, and two type IV fractures. The fracture of tibial eminence was most often related to skiing (four cases), playing football (three cases), and falling from a bicycle (two cases). On average, our study’s surgery was 4.52 days from the initial trauma (SD 3.38). Ten patients were treated by ARIF. In the remaining seven patients, conversion to ORIF through mini-arthrotomy was required after initial arthroscopy failure. Among the patients in whom mini-arthrotomy was necessary, we diagnosed one type II, four type III, and two type IV fractures. Two bioabsorbable pins were often used to stabilize the avulsed eminence (13 patients; SD 0.63). The mean duration of the surgery in our study was 81 min (SD 14.25). In one patient, we diagnosed an accompanying partial ACL lesion (patient no. 10). After treatment with the use of bioabsorbable pins, we obtained a very high healing rate—17 patients with the complete union on the control radiographs (including two cases of malunion—patients no. 2 and 10). The imaging studies of patient no. 10 are presented in Figure 4. At follow-up, we diagnosed four patients with a limited range of motion in the affected knee. Two senior orthopedic surgeons performed a clinical examination of knee joint stability independently. Only one patient showed a slight positive pivot shift test (grade II) with a negative Lachman test and negative anterior drawer test (patient no. 10). The treatment effect at follow-up was assessed using the Lysholm Knee Score and IKDC Score. The median Lysholm Score was 96.64 (SD 4.54), and the median IKDC Score was 84.64 (SD 3.10), which were both excellent results. After treatment, 14 patients returned to sports activities at least at the pre-trauma level. There was no postoperative infection or synovitis in any of the operated knees. We did not recognize any clinical signs of growth disturbance regarding leg-length discrepancy. Furthermore, we did not observe any proximal tibia growth arrest or hyperextension greater than five degrees, specified as knee recurvatum.

The above-stated results are summarized in Table 2.

## 4. Discussion

Because bioresorbable implants are rapidly growing alternatives to traditional orthopedic devices, we aimed to evaluate the usefulness of these implants in treating tibial eminence fractures among the pediatric population. The main goal of treatment is to attain fracture union and, thanks to this, to restore knee stability, shorten the time of postoperative immobilization, and bring back the full range of knee motion. Based on our results, surgery using bioabsorbable devices for type II, III, and IV of tibial eminence fractures in young individuals is an effective alternative, allowing good outcomes and restoring proper knee stability. Moreover, the omission of a second surgery for implant removal is beneficial, specifically in juveniles.

### 4.1. Surgical or Non-Surgical Treatment

Over the years, we have observed changes in tibial eminence fractures’ clinical approaches and treatment philosophies. Generally, many papers published before 2000 opted for conservative treatment. Wilfinger C et al., in 2009, presented 43 patients (including 14 type I, 13 type II, and 16 type III fractures) who underwent only closed reduction with no further internal fixation [25]. Based on their findings, they postulated that there is no reason for tibial eminence fixation. Only one patient required reoperation because of a non-union. Interestingly none of the patients reported giving way. Hunter RE et al. recommended surgical treatment only for type II fractures that did not reduce in extension [16]. Moreover, at this point, the possibility of breaking a type II fracture into a type III during a closed reduction attempt is worth emphasizing [6]. Recently, surgical reduction and internal fixation for type II, III, and IV tibial eminence fractures have been recommended by many authors, and have become a gold standard [13,26,27,28,29].

### 4.2. Surgical Treatment: Arthroscopic or Open Reduction

Arthroscopic treatment is becoming more popular nowadays, and it is related to earlier postoperative mobilization and shorter hospital stays. Considering specific fractures of the tibial eminence, the anterior horn of the medial meniscus or the transverse ligament could interpose between the fracture and constitute a significant problem during tibial eminence reduction, especially by arthroscopy [30,31]. Type IV of tibial eminence fracture could be another limitation of arthroscopic treatment because it is difficult to make an anatomical reposition through arthroscopic ports. In such cases, ARIF may be associated with a prolonged surgery duration, which increases the risk of possible complications.

Diagnostic knee arthroscopy was the fundamental procedure for all our patients. In case of arthroscopic approach failure, mini-arthrotomy has been used for direct visualization and optimal fracture reduction and further fixation. In our study group, seven patients underwent the ORIF procedure (type II—one case; type III—four cases; type IV—two cases) and ten the ARIF procedure. The key was to avoid elongating the arthroscopy and to proceed with stable fracture fixation. Therefore, the number of ORIF procedures in our group was more significant than in other studies. After performing multivariate analysis, Watts CD et al. [32] found that operative time ≥ 120 min is a risk factor for postoperative difficulties. Our study’s mean duration of surgery was 81 min (SD 14.25), which probably corresponds with the low number of complications.

As well as the conclusions of Watts CD et al. [32], both ARIF and ORIF are acceptable methods of treatment, and the system of decision-making should depend on the surgeon’s experience and the type of fracture. ACL-equivalent injuries should be managed efficiently, so it is crucial to make a decision of mini-arthrotomy extension with no delay and a prolonged arthroscopic approach.

### 4.3. Surgical Treatment, Implant Selection

Reviewing the literature, there is no consensus about which type of fixation is the best. Most papers are clinical series, and there are no randomized controlled trials that compare particular treatment methods. Many different stabilizations are known. The most popular are K-wires, metal screws, and sutures. Surgical reduction with K-wire fixation gives good results, with a low rate of knee instability. The risk of physeal arrest using this technique is low, which can make it beneficial for children. Moreover, K-wires can be removed in the outpatient clinic, which is an advantage, but otherwise, it can be related to higher infection risk [33,34]. Many authors use many kinds of metal screws for fixation. However, these devices are supposed to be removed due to the risk of eventual cartilage damage, knee impingement, or tibial growth arrest [10,28,35,36,37]. Headless Herbert screws can reduce the above-mentioned risk [38]. Some papers advocate that suture fixation can be optimal for comminuted fractures, decreasing the risk of further comminution. However, they have lower mechanical resistance [14,39,40,41].

The implants which were used in our study are bioabsorbable bone fixation nails designed for osteochondral fragment fixation [18,19]. A surgical technique in which the osteochondral fragments are stabilized with biodegradable pins allows the omission of metal implant insertion, which is important for intra-articular pediatric fractures. Bioabsorbable pins do not need subsequent removal. Nails have appropriate initial mechanical strength and stiffness, which gives the possibility of adequate fracture stabilization. The protrusions on the nail body allow for a stable implant anchoring in the bone canal [15,19]. However, the pull-out strength for SmartNail was estimated at only 61 N, and it could be related to a potential destabilization, but we did not observe such a complication in our study [15]. Excellent results of treating tibial eminence fracture with the use of bioabsorbable SmartNails are consistent with the results published by Liljeros K et al. [13]. They treated 13 patients with the ARIF approach, although no type IV fractures were included in their study. Similar to our results, they found slight anterior knee instability in only one patient, but they found a minor range of motion restrictions in six patients (four patients in our study). In contrast to our study, in Liljeros K et al. group, three to four nails were used in each patient (an average of two in our group), and the operated knee was immobilized in slight flexion for five weeks (no longer than four weeks in our group). They were forced to convert the arthroscopy into an open procedure because of technical problems only in one case (fracture type III).

### 4.4. Complication

The most common complications of the tibial eminence fracture are non-union or mal-union, anterior knee pain, knee laxity, and arthrofibrosis [32,42,43]. Arthrofibrosis is the most severe complication. Delayed surgery (≥7 days from initial injury) and prolonged postoperative cast immobilization (≥28 days) have been recognized as risk factors [32,42]. Patel et al. after-retrospective review of 40 patients treated with closed reduction stated that those who had immobilization for more than four weeks were 12 times more likely to develop arthrofibrosis than those who started rehabilitation before four weeks [42]. Nevertheless, numerous available postoperative protocols recommend knee immobilization in extension for four to six weeks [10,44].

In our study group, the surgical procedure was performed on average 4.5 (ranging from 1 to 10) days after the initial trauma. For our patients, the same protocol of postoperative immobilization in a splint for four weeks was followed with subsequent hinge orthosis replacement. Despite the above, we did not observe any case of arthrofibrosis in our group. None of our patients needed any surgical manipulation under anesthesia and/or lysis of joint adhesions. In contrast to Smith et al. [45], who postulated that even though the fracture heals in its anatomical position, due to the ligament stretching, mild degrees of anterior cruciate ligament laxity often occur. In our study, we recognized mild anterior knee instability only in one patient.

We found one case of malunion (case no. 2) but interestingly, not related to any pain or knee giving way, with a flexion deficit of 10 degrees. It was related to the solid healing of tibial ACL attachment despite the elevation of the anterior eminence portion by 3 mm. One patient with malunion with concomitant partial ACL lesion (case no. 10) required secondary arthroscopy four years after the first operation due to the impingement in the anterior knee compartment related to a 10-degree extension deficit (MRI and CT examination revealed mal-union of the tibial eminence with the elevation of its front edge of 6 mm—Figure 3). Shaving of bone eminence combined with a partial synovectomy (hypertrophied synovium) gave a very good effect, including regaining full knee extension. Due to the absence of clinical and subjective symptoms of knee instability, the patient resumed soccer training, but she remains under control.

### 4.5. Study Limitations

The study has some limitations. The most important is retrospective, observational design. Small samples, lack of a direct control group, and short-term follow-up are also restrictions. Moreover, anterior knee instability was not evaluated with a knee arthrometer, which is a limitation. There was an insufficient sample size for statistical measurements.

## 5. Conclusions

Considering tibial eminence fractures in pediatric orthopedic care, both ARIF and ORIF are acceptable, and the decision-making system should always depend on the surgeon’s experience and the fracture type. Based on the results, surgery using bioabsorbable devices for types II, III, and IV of tibial eminence fractures in young individuals is an effective alternative, allowing for good outcomes and restoring proper knee stability. The crucial factor for a good effect is a stable fracture fixation. Arthroscopic surgery gives good outcomes with minimal invasion. It is important not to prolong the attempts of arthroscopic reduction, and to perform the open reduction to shorten the procedure’s time and avoid complications.

## Figures and Tables

**Figure 1 children-09-01897-f001:**
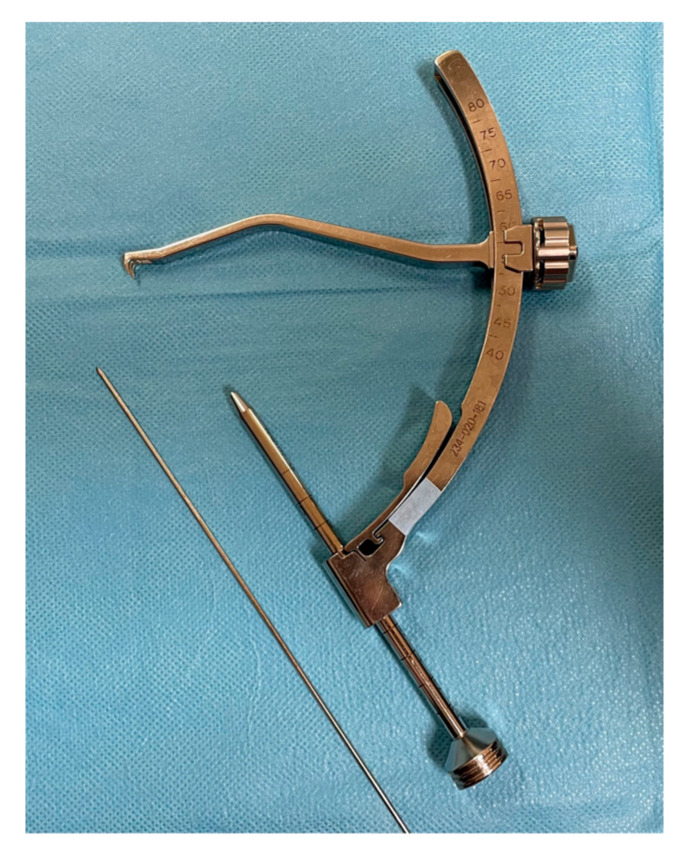
Sharp-ended tibial guide from ACL reconstruction set used to reduce and to stabilize tibial eminence with K-wire by retrograde drilling. A sharp tip is inserted through the anteromedial arthroscopic portal into the joint on top of the fragment to reduce and hold the fracture in the desired position. After establishing the desired alignment, a fracture is fixed by retrograde drilling 1.2 mm K-wire through the assembled tube.

**Figure 2 children-09-01897-f002:**
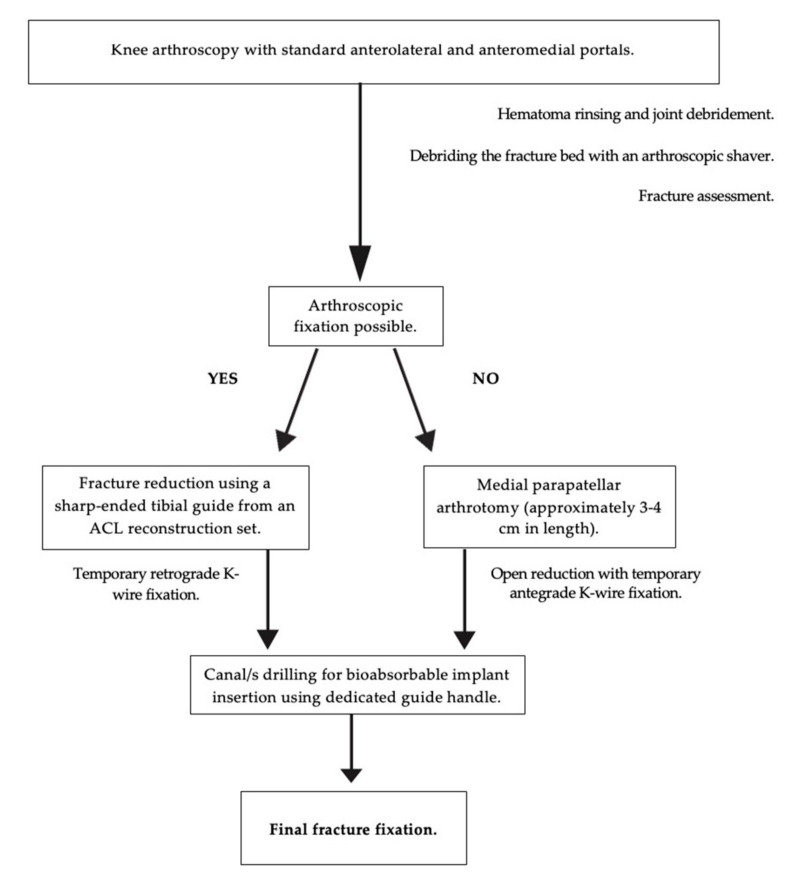
The diagram of the surgical procedure scheme.

**Figure 3 children-09-01897-f003:**
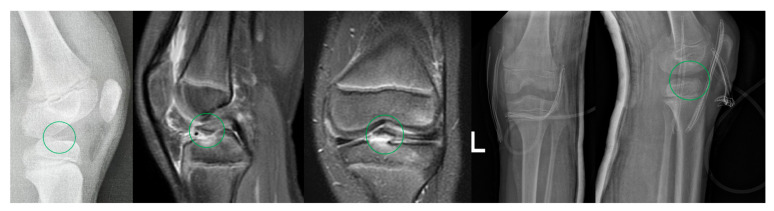
9-year-old boy with a left (L) tibial eminence fracture. Chronological: lateral X-ray after injury, sagittal and coronal MRI scans before surgery, AP, and lateral X-rays one day after ARIF procedure (fixation with 2 SmartNails). Tibial eminences were marked.

**Figure 4 children-09-01897-f004:**
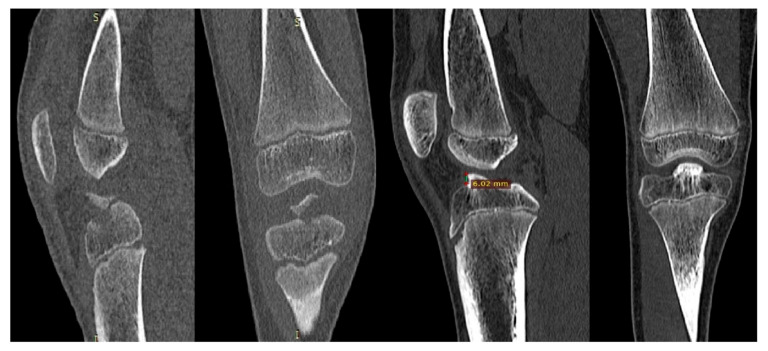
A 7.3-year-old girl with a right tibial eminence fracture-type IV according to Meyers–McKeever classification. Chronological: CT after injury, sagittal and coronal scans before surgery; CT before second surgery, sagittal and coronal scans (malunion of the tibial eminence with the elevation of its front edge of 6 mm).

**Table 1 children-09-01897-t001:** Study group overview. BMI—Body Mass Index, L—left, R—right, ARIF—arthroscopic reduction and internal fixation, ORIF—open reduction and internal fixation, SN—SmartNail.

No.	Sex	Age	BMI	Mechanism of Injury	Meyers-McKeever Classification. Site	Time to Surgery [Day]	Surgery Type. Number of Bioabsorbable Pins	Accompanying Trauma.	Follow-Up [Month].
1	Boy	13	17.5	Football match	III/L	4	ORIF–2× SN		24
2	Girl	5, 3	18.18	Motorcycle accident	II/R	10	ARIF–1× SN		102
3	Boy	10, 3	16.5	Football match	II/R	13	ORIF–2× SN		18
4	Girl	8, 5	17.5	Playground slide	III/R	5	ORIF–2× SN		22
5	Boy	10, 8	16.7	Fall from bicycle	III/L	2	ORIF–2× SN		26
6	Girl	5	14.7	Trampoline	III/R	1	ARIF–2× SN		16
7	Boy	7, 9	21.9	Fall from bicycle	III/L	2	ARIF–2× SN		30
8	Boy	12, 9	23.3	Car accident	II/R	8	ARIF–1× SN		24
9	Boy	12, 1	27	Skiing	III/L	4	ORIF–2× SN		25
10	Girl	7, 3	14.1	Fall from ladder	IV/R	5	ORIF–4× SN	Partial ACL damage	60
11	Girl	8, 3	16.1	Skiing	III/R	1	ARIF–2× SN		20
12	Boy	10, 2	17.8	Skiing	IV/R	1	ORIF–2× SN		24
13	Boy	13, 6	20.7	Physical abuse	II/L	7	ARIF–2× SN		25
14	Boy	9, 1	16.7	Fall from own height	II/L	3	ARIF–2× SN		24
15	Boy	12, 1	22.3	Football match	III/L	3	ARIF–1× SN		14
16	Boy	10, 1	20.4	Skiing	III/L	7	ARIF–2× SN		14
17	Boy	15, 2	20.3	Basketball match	III/L	1	ARIF–2× SN		9

**Table 2 children-09-01897-t002:** Individual treatment results.

No.	Radiographic Union	Extension Deficit	Flexion Deficit	Lachman Test	Drawer Test	Pivot Shift Test	Lysholm Score at Follow-Up	IKDC Score at Follow-Up	Activity Level at Follow-Up
1	Yes			/-/	/-/	/-/	100	87	Same
2	Malunion		10°	/-/	/-/	/-/	95	83	Lower
3	Yes			/-/	/-/	/-/	100	87	Same
4	Yes			/-/	/-/	/-/	100	87	Same
5	Yes			/-/	/-/	/-/	95	84	Same
6	Yes			/-/	/-/	/-/	100	87	Same
7	Yes			/-/	/-/	/-/	100	87	Same
8	Yes			/-/	/-/	/-/	100	87	Same
9	Yes			/-/	/-/	/-/	100	87	Same
10	Malunion	10°	10	/-/	/-/	II°	85	78	Lower
11	Yes			/-/	/-/	/-/	100	87	Same
12	Yes			/-/	/-/	/-/	91	79	Same
13	Yes			/-/	/-/	/-/	91	81	Same
14	Yes			/-/	/-/	/-/	100	87	Same
15	Yes	5°		/-/	/-/	/-/	95	81	Lower
16	Yes	5°		/-/	/-/	/-/	91	83	Same
17	Yes			/-/	/-/	/-/	100	87	Same

## Data Availability

The data generated and analysed in the current study are available from the corresponding author on reasonable request.

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
