# Peer review of "Results of Anterior Cruciate Ligament Avulsion Fracture by Treatment Using Bioabsorbable Nails in Children and Adolescents"

_children, 2022, doi:10.3390/children9121897_

Round 1
Reviewer 1 Report
Overall, it is considered to be a well-written paper with a good structure as a clinical study of an interesting subject.
However, it seems that it is necessary to correct the expression in general and the specific part.
Please edit the title a little more to fit the grammar.
eminence fracture ? or ACL avulsion fracture?
Abstract
line 18: Numbers should be displayed to the first decimal place.
line 20: control X-ray? please modify
line 25: ARIF, ORIF? (Where is your full name?)
Introduction
There is no need to lengthen the description of classification in the first paragraph and in Table 1, please delete table 1.
In Introduction, the authors explain the advantages and disadvantages of surgery using the materials used in surgical treatment of these diagnosed patients, and explain why clinicians are interested in the results of surgery using these methods. The description should flow and be written.
Considering this point, it is considered that the introduction needs to be rewritten.
Material and methods
line 65: CT (full name)
line 99: 9 years old
line 103: clinical follow-up or clinical assessment
line 108: Tegner activity level? or Lysholm score? please clarify and add citation
Result
"Meyers – McKeever classification" is repeated too much in this article. Take this part into consideration and delete some.
Line 119: Numbers should be displayed to the first decimal place.
Line 124-125: Once you've marked it, write it in an abbreviation next time.
In result section, Did you not calculate the average value of lachmann or ADT?
In addition to ACL avulsion fracture in children, knee recurvatum or malalignment can also be important outcomes. Is there any report on this?
Discussion
Line 152-157: please summarize results or findings of your study.
Remove all subtitles from Discussion.
line 168: edit "emphasizing"
line 170: gold standard? It's too much.
line 209: avoid the use of commercial name of implant
Avoid parentheses as much as possible.
line 247: mal-union -> malunion
Conclusion
line 264-265: remove this sentence. It's too subjective.
Author Response
Dear reviewer.
We would like to kindly thank you for your time spent reviewing our manuscript ‘’ Results of anterior cruciate ligament avulsion fracture treatment using bioabsorbable nails in children and adolescents’’. We appreciate all your valuable comments of our work. We would like to emphasize that all revisions made were marked up using the “Track Changes” function in MS Word so changes can be easily viewed.
Best regards.
Łukasz Wiktor
-
eminence fracture ? or ACL avulsion fracture? - we changed the title to: "Results of anterior cruciate ligament avulsion fracture treatment using bioabsorbable nails in children and adolescents".
Abstract
-
line 18: Numbers should be displayed to the first decimal place - we made a correction.
-
line 20: control X-ray? please modify - we changed to control radiographs.
-
line 25: ARIF, ORIF? (Where is your full name?) – we re-written conclusions.
Introduction
-
There is no need to lengthen the description of classification in the first paragraph and in Table 1, please delete table 1 - we removed table 1 as recommended. In our opinion, the description of the classification should be left in the introduction section as it is not widely known in the orthopedic community dealing with adults exclusively.
Material and methods
· line 65: CT (full name) - we changed to computed tomography
-
line 99: 9 years old - we made a correction.
-
line 103: clinical follow-up or clinical assessment - we changed to clinical follow-up
-
line 108: Tegner activity level? or Lysholm score? please clarify and add citation - we made a correction and added citation.
Result
-
"Meyers – McKeever classification" is repeated too much in this article. Take this part into consideration and delete some - we made a correction.
-
Line 119: Numbers should be displayed to the first decimal place - we made a correction.
-
Line 124-125: Once you've marked it, write it in an abbreviation next time - we made a correction (10 patients were treated by ARIF. In the remaining 7 patients, conversion to ORIF through mini-arthrotomy was required after initial arthroscopy failure).
-
In result section, Did you not calculate the average value of lachmann or ADT? – we did, we added data and citation. (A comprehensive knee examination was performed for all patients, including an anterior drawer test, Lachman test, and pivot shift test compared with the unaffected knee. We estimated the results of the Lachman test and anterior drawer test if positive as grade I with an anterior tibial translation of 1- 5 mm, grade II with a translation of 6-10 mm, and grade III with laxity more than 10 mm. For the pivot-shift test, we assumed grade 0 as normal, with no reduction or shifting, grade I with mild sliding, grade II with moderate tibial shifting while reducing, and grade III with the tibia starting subluxed and reducing with flexion, causing a characteristic clunk).
-
In addition to ACL avulsion fracture in children, knee recurvatum or malalignment can also be important outcomes. Is there any report on this? - we did not recognize any clinical signs of growth disturbance regarding leg-length discrepancy. Furthermore, we did not observe any proximal tibia growth arrest or any case of knee recurvatum in our study.
Discussion
-
Line 152-157: please summarize results or findings of your study – we summarized (based on our results, surgery using bioabsorbable devices for type II, III, and IV of tibial eminence fractures in young individuals is an effective alternative allowing good outcomes and restoring proper knee stability. Moreover, the omission of a second surgery for implant removal is beneficial, specifically in juveniles).
-
line 168: edit "emphasizing" – we changed (Moreover, at this point, the possibility of breaking a type II fracture into a type III during a closed reduction attempt is worth underlining [6]. Recently, surgical reduction and internal fixation for type II, III, and IV tibial eminence fractures have been recommended by many authors and have become a benchmark [13,24-27].
-
line 170: gold standard? It's too much - as above.
-
line 209: avoid the use of commercial name of implant – we changed to the bioabsorbable implants
-
line 247: mal-union -> malunion
Conclusion
-
line 264-265: remove this sentence. It's too subjective – we removed as recommended.
Reviewer 2 Report
General Comments
Thank you for the opportunity to review this article. This manuscript was investigated the results of tibial eminence fracture fixation with bioabsorbable nails in children and adolescents. I read this manuscript with interesting. As you also mentioned, treatment of tibial eminence fracture is interesting problem for knee surgeons. You described good outcomes of tibial eminence fracture fixation with bioabsorbable nails. However, I have some questions that need to be improved in this manuscript. First, “Surgical Technique” is difficult for readers to understand. Authors should describe surgical technique in detail and add more figures including operative pictures. Second, conclusion is not matched to the purpose and result in this study. Conclusion should be written more concisely. And conclusion in text should be matched to abstract.
I feel this manuscript reaches the qualifications for publication If authors make major changes.
Specific Comments:
Abstract
Introduction
Please see the general comments.
Materials and Methods
Line 58-66
MRI was not use for diagnosis in this study? I think MRI is useful to diagnose the concomitant injury such as meniscus tear.
Results
Line 120
Is “fife” spell mistake? If so, “fife” should be corrected “five”.
Line 124
“ARIF” is defined in line 47.
Line 125
“ORIF” is defined in line 49.
Line 132-133
Two cases were malunion in abstract and Table 3. However, line 132-133 was described as “malalignment”. Which is correct?
Discussion
Line 182-183
Number of patients is not correct.
“Type Ⅱ-1 case; type Ⅲ-2cases; type Ⅳ-2 cases” were 5 cases!
Line 209-210
Please add the citation.
Line 259-260
I think short term follow-up is also limitation in this study.
Conclusions
Please see the general comments.
Author Response
Dear reviewer.
We would like to kindly thank you for your time spent reviewing our manuscript ‘’ Results of anterior cruciate ligament avulsion fracture treatment using bioabsorbable nails in children and adolescents’’. We appreciate all your valuable comments of our work. We would like to emphasize that all revisions made were marked up using the “Track Changes” function in MS Word so changes can be easily viewed.
Best regards.
Łukasz Wiktor
First, “Surgical Technique” is difficult for readers to understand. Authors should describe surgical technique in detail and add more figures including operative pictures - we changed the "Surgical Technique" to make it easier for the reader to understand. We added some details in the surgery description. We attached a diagram of the medical procedure (figure 2) to make it more transparent. We completed the description under figure 1. Unfortunately, we do not have valuable intraoperative photos, but we hope this will not be why the manuscript can be rejected.
Second, conclusion is not matched to the purpose and result in this study. Conclusion should be written more concisely. And conclusion in text should be matched to abstract - we changed the conclusions section to correspond to the study's results.
Specific Comments:
Abstract
Introduction
Please see the general comments.
Materials and Methods
-
Line 58-66 MRI was not use for diagnosis in this study? I think MRI is useful to diagnose the concomitant injury such as meniscus tear - we added comment in article (Additional magnetic resonance imaging (MRI) was done only on a few patients before the surgery in our study. We believe MRI can also be helpful, especially in diagnosing concomitant knee injuries such as meniscus tears). Since our diagnosis was based mainly on CT and MRI performed on only a few patients, we did not include MRI results in our study.
Results
-
Line 120 Is “fife” spell mistake? If so, “fife” should be corrected “five” - we made a change.
-
Line 124 “ARIF” is defined in line 47 - we removed unnecessary explanation.
-
Line 125 “ORIF” is defined in line 49 - we removed unnecessary explanation.
-
Line 132-133 Two cases were malunion in abstract and Table 3. However, line 132-133 was described as “malalignment”. Which is correct? - malunion is correct one, we made change.
Discussion
-
Line 182-183 Number of patients is not correct. “Type Ⅱ-1 case; type Ⅲ-2cases; type Ⅳ-2 cases” were 5 cases! - Type Ⅱ-1 case; type Ⅲ-4cases; type Ⅳ-2 cases is the correct version, we made change.
-
Line 209-210 Please add the citation - we added citations.
-
Line 259-260 I think short term follow-up is also limitation in this study - we added recommended limitation (The study has some limitations. The most important is retrospective, observational design. Small samples, lack of a direct control group, and short-term follow-up are also restrictions).
Conclusions
Please see the general comments.
Round 2
Reviewer 1 Report
It was confirmed that most of the points we recommend were satisfactorily corrected. But still there are some points that should be improved for publication.
Title: please add "by" before treatment.
Abstract
line 14: please modify "typical".
line 17: please modify" orthopaedic center", is it orthopaedic clinic? or general hospital?
line 18: remove "our"
line 19: please specify " bioabsorbable implants" is it pin? or nail?
line 22: very high? please minimize subjective expression.
line 23: second -> revision
line 29: please modify the sentence "stable fracture...".
line 49: recheck "Keywords" you can add more keywords for this paper.
Introduction
: same expression or sentence with abstract are required to be modified by my recommendation as above.
line 63: correct ACL -> correction of ACL
line 64: please rewrite the sentence. that is so confused.
line 66: please use same fashion to write the "ARIF" for brief letters.
ex. arthroscopic reduction and internal fixation (ARIF)
line 69: delete "many"
please add the weak points of bioabsorbable nail or clinical concerns which many clinician hesitate to use this material...ex.. weak strength or invisible in follow up x-ray evaluation.
2. M&M
line 105: please modify "yo"
line 110: please write full name of CT
line 113: please avoid subjective expression. helpful seems little bit subjective.
Table 1. : please rewrite the title of table.
and move the details about table below the table.
line 130: finland? Conmed? isn't it US company?
line 150,151: remove "desire"
line 157: what is the "splint cast"?
line 159: please recheck English grammer "rehabilitation's".
line 193: please rewrite the title for figure 2.
line 203-204: recheck English grammer.
line 212: I think the Lysholm score is scoring system for functional outcome, not subjective outcome.
line 214-216: please recheck the grammer and avoid parentheses.
3. Results
please add standard deviation or standard error at every numbers in result section.
line 259-262: what is the definition of "knee recurvatum: of your study? please add it.
line 289-290: please avoid "-"
line 294: please rewrite the title of Table 3.
Discussion
line 297: implant's ? please check the grammer.
line 305: non-surgical -> conservative
line 306-307: please avoid subjective opinion or recheck the grammer.
line 308-309: please rewrite the sentence following English grammer.
line 319: more and more?
line 332-335: please rewrite the sentence following English grammer.
line 363: please check the first sentence.
line 367-369: please add reference.
line 389: Patel et al. seems better.
line 395: please remove "strict"
line 417-419: is it all for limitation? please add more things about weakness of your study.
Conclusion
recheck the grammer and expressions in the sentences..
Author Response
Dear reviewer.
We would like to kindly thank you for reviewing our manuscript. We appreciate all your valuable comments of our work. We would like to emphasize that all revisions made were marked up using the “Track Changes” function in MS Word so changes can be easily viewed.
Best regards.
Łukasz Wiktor
Title
Title: please add "by" before treatment - we made the corrections.
Abstract
line 14: please modify "typical" - we made the corrections.
line 17: please modify" orthopaedic center", is it orthopaedic clinic? or general hospital- we made the corrections.
line 18: remove "our"- we made the corrections.
line 19: please specify " bioabsorbable implants" is it pin? or nail? - we made the corrections.
line 22: very high? please minimize subjective expression - we made the corrections.
line 23: second -> revision - we made the corrections.
line 29: please modify the sentence "stable fracture...". - we made the corrections.
line 49: recheck "Keywords" you can add more keywords for this paper - we added keywords.
Introduction
line 63: correct ACL -> correction of ACL - we made the corrections.
line 64: please rewrite the sentence. that is so confused - we corrected sentence.
line 66: please use same fashion to write the "ARIF" for brief letters ex. arthroscopic reduction and internal fixation (ARIF) - we made the corrections.
line 69: delete "many" – we made the corrections.
please add the weak points of bioabsorbable nail or clinical concerns which many clinician hesitate to use this material...ex.. weak strength or invisible in follow up x-ray evaluation - we added limitations.
- M&M
line 105: please modify "yo"- we made the corrections.
line 110: please write full name of CT- we made the corrections.
line 113: please avoid subjective expression. helpful seems little bit subjective- we made the corrections.
Table 1. : please rewrite the title of table. and move the details about table below the table -we made the corrections.
line 130: finland? Conmed? isn't it US company? - We have implants that are manufactured in Finland: SmartNails (ConMed, Linvatec, Finland).
line 150,151: remove "desire"- we made the corrections.
line 157: what is the "splint cast"? - we made the corrections.
line 159: please recheck English grammer "rehabilitation's"- we made the corrections.
line 193: please rewrite the title for figure 2- we made the corrections.
line 203-204: recheck English grammer- we made the corrections.
line 212: I think the Lysholm score is scoring system for functional outcome, not subjective outcome- we made the corrections.
line 214-216: please recheck the grammer and avoid parentheses- we made the corrections.
- Results
please add standard deviation or standard error at every numbers in result section – we added data.
line 259-262: what is the definition of "knee recurvatum: of your study? please add it – we added definition.
line 289-290: please avoid "-"- we made the corrections.
line 294: please rewrite the title of Table 3- we made the corrections.
- Discussion
line 297: implant's ? please check the grammer- we made the corrections.
line 305: non-surgical -> conservative- we made the corrections.
line 306-307: please avoid subjective opinion or recheck the grammer- we made the corrections.
line 308-309: please rewrite the sentence following English grammer- we made the corrections.
line 319: more and more? - we made the corrections.
line 332-335: please rewrite the sentence following English grammer- we made the corrections.
line 363: please check the first sentence- we made the corrections.
line 367-369: please add reference- we added reference.
line 389: Patel et al. seems better. - we made the corrections.
line 395: please remove "strict"- we made the corrections.
line 417-419: is it all for limitation? please add more things about weakness of your study- we added limitations.
- Conclusion
recheck the grammer and expressions in the sentences-- we made the corrections.
Reviewer 2 Report
General Comments
Thank you for giving me the opportunity to review this manuscript.
I think it has been corrected according to the review comment. Surgical Technique in Material and methods is rewritten clearly and improved. Moreover, I think Figure 2 is useful to understand this study for readers.
Author Response
Dear reviewer.
We would like to kindly thank you for reviewing our manuscript.
Best regards.
Łukasz Wiktor